# Recent Strategies for High-Performing Indoor Perovskite Photovoltaics

**DOI:** 10.3390/nano13020259

**Published:** 2023-01-07

**Authors:** Kelvian T. Mularso, Ji-Young Jeong, Gill Sang Han, Hyun Suk Jung

**Affiliations:** 1School of Advanced Materials Science and Engineering, Sungkyunkwan University, Suwon 16419, Republic of Korea; 2SKKU Institute of Energy Science and Technology (SIEST), Sungkyunkwan University, Suwon 16419, Republic of Korea

**Keywords:** hybrid organic–inorganic perovskite, wide bandgap, indoor photovoltaics, low-light illumination, internet of things, perovskite solar modules

## Abstract

The development of digital technology has made our lives more advanced as a society familiar with the Internet of Things (IoT). Solar cells are among the most promising candidates for power supply in IoT sensors. Perovskite photovoltaics (PPVs), which have already attained 25% and 40% power conversion efficiencies for outdoor and indoor light, respectively, are the best candidates for self-powered IoT system integration. In this review, we discuss recent research progress on PPVs under indoor light conditions, with a focus on device engineering to achieve high-performance indoor PPVs (Id-PPVs), including bandgap optimization and defect management. Finally, we discuss the challenges of Id-PPVs development and its interpretation as a potential research direction in the field.

## 1. Introduction

The Internet of Things (IoT) refers to a network where physical objects or devices are integrated with sensors, programs, apps, and other automation technologies so that they can be connected by exchanging/transferring data via the Internet. This is supported by the advancement of real-time cloud-based network interconnections [1], extending to advanced technologies, such as in daily life, healthcare, transportation, and manufacturing industries.

The IoT is already widely used in smart appliances [2], such as smartphones, smart TV, smart air conditioning systems, and smart lighting, as well as in the manufacturing industry [3]. Additionally, IoT sensors consist of two main components: (1) the sensors necessary in each environment, which require specific features, that are small-size, high-performing, and wireless compatible, and (2) a power source. Batteries are the most used power source for IoT sensors. However, the automation system is partially shut down when the battery capacity is low, leading to faulty control. Among the energy harvesting devices, indoor photovoltaics are one of the most promising power sources for charging batteries or operating self-powered IoT sensors. Furthermore, indoor photovoltaics have a higher power generation capacity under indoor light compared to other energy-harvesting devices, such as triboelectric and piezoelectric nanogenerators [4], mechanical energy [5], sound energy [6], radio frequency [7], and thermal energy harvesters [8]. Particularly, perovskite photovoltaics (PPVs) exhibit higher light-to-power conversion efficiency (PCE) under indoor light compared to other solar cells including silicon-based (Si) [9,10], copper indium gallium selenide (CIGS) [11], organic [12,13,14,15,16], and dye-sensitized solar cells (DSSC) [17,18,19].

Since the introduction of PPVs in 2012 [20], perovskite materials with the general formula ABX_3_, which are composed of organic (CH_3_NH_3_^+^ or MA^+^, CH(NH_2_)_2_^+^ or FA^+^), inorganic (Cs^+^), or mixed organic–inorganic A cations, intermetallic compound B (Sn^2+^, Pb^2+^, etc.), and single or combined halide X anions (Cl^−^, Br^−^, I^−^), have been extensively investigated over the past 10 years. Due to their unique optoelectronic properties, PPVs have recorded a performance of up to 25.7% of PCE for various thin-film engineering [21,22]. Additionally, the benefits of mechanical flexibility [23], low production cost (i.e., by replacing the expensive gold with carbon-based materials as an electrode, and processing under ambient conditions [24,25,26,27]), and relatively low-temperature processing capability [28] make it suitable for vast applications [29], including portable and flexible electronics, textiles-electronics integration, and transportation units, such as unmanned aerial vehicles. Furthermore, among all indoor photovoltaics, indoor perovskite photovoltaics (Id-PPVs) exhibit the highest PCEs of over 40% under indoor light [30,31], which are suitable for wireless and self-powered IoT sensors. However, there are still rooms for further improvements in Id-PPVs to achieve better stability and higher efficiency derived from the Shockley-Queisser limit under indoor light illumination [32,33].

In this review, we discuss the current progress in Id-PPVs technology with detailed strategies applied to date. This is based on state-of-the-art Id-PPVs, followed by a comprehensive explanation of perovskite and charge transport material engineering. Finally, the challenges and the prospects of Id-PPVs are discussed. This review aims to elaborate on the development tendency and further development in Id-PPVs.

## 2. Id-PPVs: Light Properties and Band Gap Optimization

Indoor lamps, such as light emitting diodes (LEDs), fluorescent lamps (FLs), and halogen lamps, have a different spectrum compared to that of artificial sunlight, which is standardized as 1 sun illumination and denoted as AM 1.5G, as shown in Figure 1a. The photovoltaic performance of a device with a certain bandgap is significantly dependent on the type of lamps used and is very sensitive to illumination intensity. Based on the calculated PCE as a function of the band gap derived from the Shockley–Queisser limit, a maximum theoretical PCE of approximately 33% under AM 1.5G is achieved for materials with a 1.20–1.40 eV bandgap [32], whereas PCEs of 45.70%, 47.70%, and 58.40% under FL, phosphorous LED, and red-green-blue LED, respectively, are achieved for materials with an optimal bandgap of 1.70–2.00 eV [33]. Figure 1b shows the PCE as a function of the bandgap for AM 1.5G and white LED. Thus, typical perovskite used in PPVs with 1.4–1.5 eV bandgap exhibits some optoelectronic loss due to bandgap mismatch with the indoor lamp spectra, which highlights the need for further bandgap engineering to attain high-efficiency Id-PPVs.

For PPVs, xenon arc lamps have been used for solar light simulator equipment for almost two decades [34] due to their wide spectrum from ultraviolet to visible region, besides the diverse types of lamps used for Id-PPVs. Although many studies reported Id-PPVs under the same illumination intensity (mostly at 1000 lx), the different spectra of each lamp make no clear standardization, yielding incomparable results. For example, a report showed different PCEs at the same intensities of 200, 400, 800, and 1000 lx for cool white LED and halogen lamps [35]; moreover, Ann et al. showed all Id-PPV performances with only power density output instead of the commonly used PCE [36]. Some studies presented both PCE and power density outputs, which are preferable for reporting device performance [37,38]. In the future, the method of calibration and standardization of the Id-PPV performance measurement needs to be elucidated to obtain a clear standardization of the different spectra of each lamp.

**Figure 1 nanomaterials-13-00259-f001:**
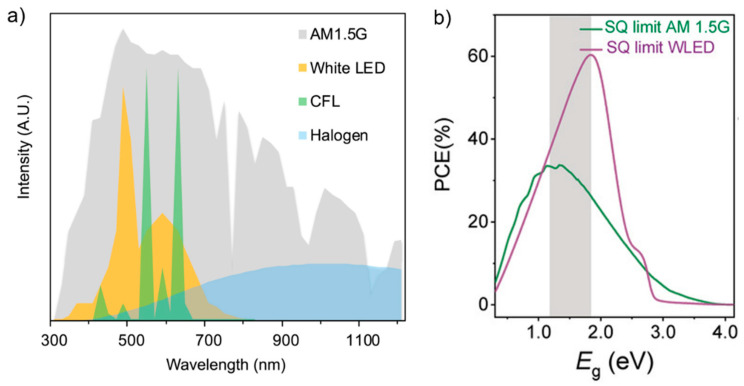
Plots depicting the (**a**) comparison of sunlight spectra (AM 1.5G) to other typical light sources. Reprinted with permission from [39], copyright 2019 Elsevier. (**b**) Energy gap vs. calculated power conversion efficiency (PCE) derived from S–Q limit of an ideal solar cell under AM 1.5G and 4000 K white light emitting diode (LED; grey area depicted different of those ideal bandgaps). Reprinted with permission from [32], copyright 2022 John Wiley and Sons.

## 3. High Performance Id-PPVs

### 3.1. Perovskite Absorber Layer

#### 3.1.1. Perovskite Bandgap Tuning

As discussed above, the highest theoretical PCE of approximately 60% for Id-PPVs is achieved by an absorber material with a bandgap of ~1.7–2.0 eV. This wide bandgap can be realized by employing a mixed-halide composition. Wu et al. used bromide additives in iodide-based perovskite films to alter the bandgap [40]. Cells with bromide fractions of 0, 0.15, 0.25, and 0.35 exhibited bandgaps of 1.59, 1.66, 1.71, and 1.78 eV, respectively, which were determined from the onset of the absorption edge of ultraviolet–visible (UV–vis) absorption spectra, as shown in Figure 2a. A higher bromide fraction results in a higher bandgap, indicating that an ideal bandgap can be achieved. The optimal bromide-fraction of 0.15 resulted in a PCE of 26.38% under a fluorescent tube lamp at 1000 lx, with current density–voltage (J–V) curves under various light intensities, as shown in Figure 2b.

Bromide doping into iodide-based perovskites not only affects the bandgap but also influences the perovskite film morphology, either by improving crystallization kinetics [41] or by minimizing phase segregation [42]. Bromide and iodide exchange on perovskite crystal can induce crystal structure transformation [43], as a higher content of bromide in pure MAPbI_3_ changed the initial structure from tetragonal to pseudo-cubic (MAPbI_2_Br and MAPbIBr_2_) to cubic (MAPbBr_3_). The exchange of larger I^−^ (radii = 2.2 Å) with smaller Br^−^ (radii = 1.96 Å) while maintaining their corner-sharing affinity resulted in a slight rotation of the PbX_3_ octahedra along the 〈001〉 axis on the (001) plane [43]. Mixed bromide-iodide PPVs exhibit lower performance than pure iodide PPVs because of the short-circuit current (J_sc_) loss with an increase in the open-circuit voltage (V_oc_). Surprisingly, pure bromide devices exhibit the highest V_oc_ among others, which compromises their stability under continuous light illumination, even with a reduction in current density.

Triple-anion-based Id-PPVs are proven to increase photovoltaic performance with distinct roles: (1) Suppressing non-radiative recombination loss [44]: As the band gap increases, the conduction band also moves towards a higher energy level on the perovskite containing multiple anions, whereas the valence band remains at a similar level. This results in easier carrier extraction and further improves performance as high as 36.2% under a fluorescent lamp at 1000 lx, (2) eliminating bulk defects, which then affects the movement of photogenerated carriers and ions [45]: Intrinsic trap sites at the grain boundary, which limit Fermi level splitting, are minimized. Thus, more carriers are separated and collected in the charge transport layer (CTL), (3) retarding crystallization kinetics and forming higher crystallinity, where partial Cl^−^ substitution shrinks the lattice structure, leading to a larger grain size, fewer defects, and a more uniform surface [46,47]. The Id-PPVs with PCE over 30% are summarized in Table 1.

**Figure 2 nanomaterials-13-00259-f002:**
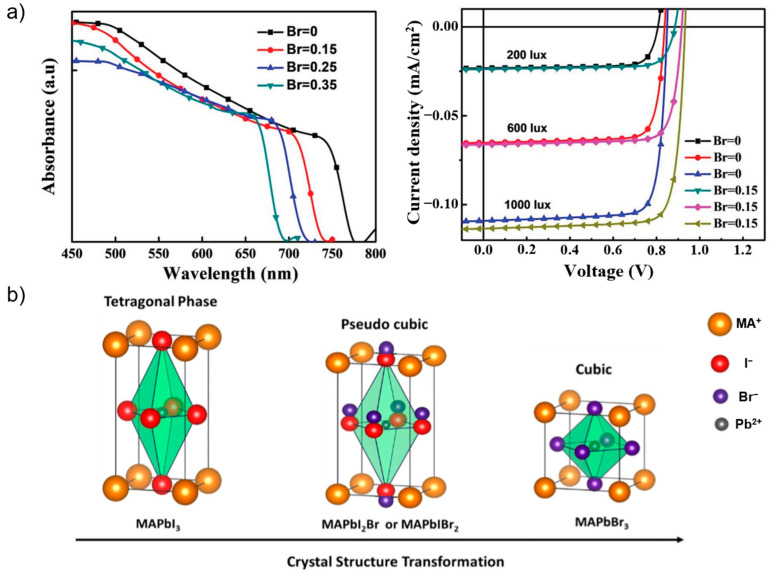
(**a**) Plots showing the absorption spectra and J–V curves of perovskite films with different Br^−^ contents (x) [MA_0.85_Cs_0.15_Pb(I_1−x_Br_x_)_3_; x = 0 or 0.15] measured under various TL5 fluorescent tube illumination. Reprinted with permission from [40], copyright 2019 John Wiley and Sons. (**b**) Schematic crystal structure transformations with different halide configurations. Reprinted with permission from [43], copyright 2021 Elsevier.

#### 3.1.2. Perovskite Crystallization Kinetics

A highly crystalline perovskite film with a low defect density is required to obtain the best performance for Id-PPVs, which can be achieved by controlling crystallization kinetics. Additive engineering is one of the preferred approaches for this purpose due to its beneficial features such as larger grain size formation, defect passivation, ion migration suppression at grain boundaries, and oxygen and moisture protection [48]. However, to date, only a few studies have incorporated additives into perovskite films. For instance, Dong et al. studied the effects of a small amount of non-halide lead oxalate additives on retarding perovskite film nucleation to induce larger grains [49]. They used in situ x-ray diffraction measurements to determine the nucleation process related to the phase and growth of the crystal. The doped film showed a slower transformation towards the black alpha phase relative to the pristine devices. The anion replacement of C_2_O_4_^−^, with a larger radius, and I^−^, with a smaller radius, immediately before perovskite phase transformation during annealing slowed down the formation of the perovskite tetragonal crystal lattice. Subsequently, they achieved a PCE of 34.86%; however, only a PCE of 31.03% was achieved for the control sample under 1000 lx illumination.

**Table 1 nanomaterials-13-00259-t001:** Summary of selected studies with power conversion efficiency (PCE) higher than 30% under low-light illumination.

Device	Perovskite Material	Key Technology	Bandgap (eV)	Intensity (lx)	Power Output (μWcm^−2^)	PCE (%)	Ref
n-i-p	FA_0.90_MA_0.05_Cs_0.05_PbI_2.85_Br_0.15_	Optimized annealing method on flexible	1.55 *	1000 ^2^	-	41.23	[50]
n-i-p	Cs_3_Sb_2_Cl_x_I_9-x_	Bionic perovskite passivation	1.59 *	1000 ^5^	-	40.24	[31]
n-i-p	(FAPbI_3_)_0.97_(MAPbBr_3_)_0.03_	Bulk and surface defect passivation	1.59	824.5 ^3^	-	40.1	[30]
p-i-n	MAPbI_3_	Modified HTL	1.50 ^#^	1000 ^3^	-	39.2	[37]
n-i-p	(MA_0.91_FA_0.09_)Pb(I_0.94_Br_0.06_)	Graft-type polymer HTL	1.60	1000 ^5^	-	38.2	[51]
n-i-p	Cs_0.05_(FA_0.85_MA_0.15_)_0.95_ Pb(I_0.85_Br_0.15_)_3_	Electron acceptor perovskite additives	-	1000 ^5^	-	37.9	[52]
n-i-p	CsPbI_2_Br	Ionic liquid perovskite additives	1.72	1000 ^2^	-	37.24	[53]
p-i-n	FA_0.8_Cs_0.2_Pb(I_0.6_Br_0.4_)_3_	Bulk and surface defect passivation	1.77	1000 ^4^	-	37.18	[32]
p-i-n	CH_3_NH_3_PbI_2−x_BrCl_x_	Triple anion perovskite	1.80	1000 ^5^	-	36.2	[44]
n-i-p	MAPbI_3_	Heterojunction ETL	-	1000	111.2	35.9	[38]
p-i-n	Cs_0.05_(FA_0.88_MA_0.12_)_0.95_PbI_3_	Alkali fluoride perovskite passivation	1.55	1000 ^4^	292	35.7	[54]
n-i-p	(FA_0.6_MA_0.4_)_0.9_Cs_0_·_1_Pb(I_0_·_6_Br0.4)3	PEA-halide additives	1.75	1000 ^4^	99.5	35.6	[55]
p-i-n	MAPbI_3_	Ionic liquid for ETL modification	1.59 *	1000 ^5^	-	35.2	[56]
n-i-p	MAPbI_3_	Non-halide perovskite additives	1.60	1000 ^5^	-	34.86	[49]
n-i-p	MAPbI_3_	Bromine-doped perovskite	1.68 *	1000 ^1^	-	34.5	[41]
n-i-p	MAPbI_3_	Blade coating	1.59 *	1000 ^4^	-	33.8	[57]
n-i-p	Cs_0.05_(FA_0.6_MA_0.4_)_0.95_Pb(I_0.6_Br_0.4_)_3_	Perovskite passivation	1.75	1000 ^2^	89.26	33.42	[35]
p-i-n	MAPbI3	NiO-based HTL	-	1000 ^2^	98.4	32.88	[58]
n-i-p	CsPbI_2_Br	Dopant-free polymer HTM	1.91	1000 ^4^	97.79	32.6	[59]
n-i-p	FAPbI_3_	Grain boundary stress release	1.52	1062 ^5^	106	31.85	[60]
n-i-p	CsFAMA-based	Polymer-based perovskite passivation	1.53	1000 ^4^	-	30.73	[61]
n-i-p	MAPbI_3_	Doped SnO_2_ ETL	1.60 ^#^	285 ^4^	-	30.3	[62]
p-i-n	MAPbI_3_	Vacuum-based coating	-	1000 ^5^	94.9	30.1	[63]

^1^ General LED, ^2^ specifically cool white LED, ^3^ specifically warm white LED, ^4^ white LED, ^5^ fluorescent lamp, * calculated from PL spectra, ^#^ calculated from the band diagram.

Ionic liquids and salts with low melting temperatures are already used in PPVs devices due to their high carrier mobility and good thermal stability [64,65,66,67,68]. Du et al. added a low concentration of a novel ionic liquid, 1-ethyl-3-methylimidazolium hydrogen sulfate (EMIMHSO_4_), into the perovskite film [53]. The EMIMHSO_4_ ionic liquid separates into EMIM^+^ and HSO_4_^−^ ions, which play different roles in the perovskite film. EMIM^+^ is distributed throughout the perovskite film and acts as an electron donor, making it more electron rich, whereas HSO_4_^−^ is located at the perovskite/TiO_2_ interface and acts as an electron acceptor. Density functional theory calculations showed that HSO_4_^−^ exhibited a higher binding affinity to the iodine vacancy (V_i_) than I^−^ and Br^−^, providing stronger coordination with undercoordinated Pb^2+^ sites while simultaneously eliminating V_i_ defects. Furthermore, the strong interaction between EMIMHSO_4_ and perovskite retarded the perovskite crystal growth, resulting in a highly crystalline perovskite film. An impressive PCE of 37.24% under 1000 lx (365 μWcm^−2^) of cool white LED illumination was obtained. Using a similar mechanism, Yang et al. employed materials with strong electron acceptors, i.e., N,N′-bis(dimethylaminopropyl-N′′′-oxide)-perylene-3,4,9,10-tetracarboxidiimide, which is a perylene diimide derivative with a terminal substituent amino N-oxide, in the perovskite active layer [52]. This facilitated faster exciton separation and transport, which subsequently yielded 37.9% of PCE under 1000 lx LED illumination compared to only 34.2% for the control samples.

In this chapter, we discussed perovskite absorber engineering for high-performing Id-PPVs, such as bandgap, additives, anion doping, and passivation. However, the formation of indoor perovskite film is sensitive with increasing bandgap due to decreasing the enthalpy of formation. To improve the crystallinity and morphology of perovskite film, it needs to further investigate the rational passivation materials, additives, and processes.

### 3.2. Charge Transport Layer

#### 3.2.1. Electron Transport Layer

The photogenerated excitons in the perovskite active layer, including electron carriers in the electron transport layer (ETL) and hole carriers in the hole transport layer (HTL), should be dissociated before collection. In the normal architecture of an n-i-p structure, it is necessary to have high optical transmittance and low electrical series resistance of the ETL. Many different types of ETL were incorporated into PPVs, such as metal-oxide materials or conductive polymers. TiO_2_ is one of the commonly used ETLs with various structures, such as mesoporous TiO_2_ (meso- TiO_2_) [69], compact TiO_2_ (c-TiO_2_) [70,71], or meso-TiO_2_ on c-TiO_2_ [72]. In Id-PPVs, meso-TiO_2_ and c-TiO_2_ are compared to organic phenyl-C_61_-butyric acid methyl ester (PCBM) [36]. At 1 sun illumination, meso-TiO_2_-based devices obtained the best performance among all, whereas c-TiO_2_-based devices were remarkable under low-light illumination (200–1600 lx). Meso-TiO_2_-based devices suffered from ideality factor changes upon low-light intensity as higher interfacial traps were observed, whereas at higher light intensities, the traps were filled by excess carriers (photodoping). The PCBM-based devices satisfied both meso- and c-TiO_2_ requirements of morphological structure and low trap density, resulting in the highest power density output under indoor light illumination.

The highest PCE reported for PPVs was achieved using SnO_2_ instead of TiO_2_, either via spin coating [21] or chemical bath deposition [22]. The SnO_2_ ETL was used, because it has favorable characteristics such as better optical and electrical properties, preferred band alignment with different types of perovskite, and higher photostability than the TiO_2_ ETL [73]. The bilayer ETL of SnO_2_/ZnO on Id-PPVs showed better film quality, lower trap density, and minimized charge recombination at the ETL/perovskite interface than single ETL, leading to low energy loss with a PCE of as much as 37.2% under 1000 lx LED illumination [24]. However, a bilayer ETL may not be a better option, because it requires an additional deposition step; thus, doping the SnO_2_ ETL can be an alternative. Bi et al. suggested that doping PbO into SnO_2_ could increase the shunt resistance and avoid leakage of the photogenerated current [62]. This is important because the photogenerated current may be similar to the shunt current at low-light illumination, which can alter the photovoltage signal, as shown in Figure 3a. Thus, a higher shunt resistance is favorable. With this strategy, 34.2% of PCE at 0.1 mWcm^−2^ illumination was obtained as compared to only 18.8% for the control samples.

#### 3.2.2. HTL

The HTL is employed in Id-PPVs to block electrons and collect holes simultaneously to achieve the best performance. Spiro-OMeTAD is a well-known HTL that is widely used for normal n-i-p structures, whereas NiO_x_ is employed for inverted p-i-n structures. To fabricate an HTL with roll-to-roll manufacturing capability on a polymer-based flexible substrate, Jagadamma et al. investigated the role of low-temperature solution-processed NiO as an HTL for inverted Id-PPVs [74,75,76,77,78,79,80]. The as-synthesized NiO HTL performed better than conventional poly(3,4-ethylenedioxythiophene) polystyrene sulfonate (PEDOT:PSS) HTL, as the hole extraction performance was better on NiO-based devices. Subsequently, a 23.0% PCE was obtained under 320 μWcm^−2^ fluorescent light. Similarly, Saranin et al. explored the effect of the perovskite configuration on compact NiO and low-temperature-processed NiO nanoparticles [58]. Although differences between their performance existed under low light and AM 1.5G, due to interface recombination and current leakage, the compact NiO devices achieved a 32.88% PCE (98.40 μWcm^−2^) under 1000 lx of LED illumination.

Other studies explored new materials to enhance hole extraction in Id-PPVs, which are novel donor–acceptor–donor molecule (D-A-D) 2,3-bis(4′-(bis(4-methoxyphenyl)amino)-[1,1′-biphenyl]-4-yl)fumaronitrile (TPA-BPFN-TPA) [81], dopant-free polymer PDTDT [59], an innovative graft-type polymer consisting of a dibenzothiophene-based conjugated backbone [51], a “V” shaped of A-D-A small organic HTL [82], and a unique new dopant of nicotinamide into PEDOT:PSS [83]. Poly(triaryl amine) was modified with hydrophobic poly(3-hexylthiophene) (P3HT) to adjust the interaction between the HTL and the undercoordinated Pb^2+^, improving the physical properties and hole extraction features [37], as shown in Figure 3b. Consequently, a very high PCE of 39.2% was achieved under 1000 lx of LED illumination, with both V_oc_ and J_sc_ improvements.

There are many efforts to improve the performance of Id-PPVs by employing the novel CTL. However, although the bandgap of perovskite film was changed in Id-PPV, it is still used as a representative charge transport layer in solar cells. The charge transport properties of CTL should consider the charge carrier mobility, low-temperature processibility, and band alignment. For the charge transport layer, a novel material with optimal band alignment is required with high mobility and low-temperature processibility for high-efficiency Id-PPVs.

### 3.3. Interface Engineering

As Id-PPVs consist of many layers, surface defects from dangling bonds may occur at the interface. These defects act as recombination centers, which can reduce the photogenerated current through carrier recombination. Thus, interface passivation is considered an efficient strategy for improving the performance of PPVs, which is already explained in detail [84]. Similarly, Li et al. incorporated perovskite layer surface passivation using phenethyl-ammonium halide (PEAH, H: I, Br, Cl) treatments [55]. They found that a thin layer of PEAH had no effect on the bandgap compared to that of the pristine samples, which was 1.75 eV. However, the surface-treated films had suppressed nonradiative recombination, as seen through the photoluminescence (PL), where the blue peak shifted from 702 to 696 nm. Although there was no detailed information about the inhibition of the best nonradiative recombination suppressing effect by PEACl-treated samples, it was shown from time-resolved PL curves that carrier lifetime was highest compared to that by PEABr- and PEAI-treated samples. Inverted Id-PPVs using PEACl treatment showed suppressed phase segregation, which resulted in 35.6% PCE with a V_oc_ of 1.08 V and fill factor of 0.83, whereas the pristine sample had a PCE of only 31% with a V_oc_ of 1.01 V and fill factor of 0.79 under 1000 lx intensity of 3000 K LED. This is consistent with the findings of a study employing PEAI as a hole-selective layer, which improved the crystallinity of the perovskite film [35].

To further improve the PCE of Id-PPVs, a holistic approach combining additive and interface engineering is a promising strategy. With this strategy, an impressive 40.1% record PCE was obtained by introducing guanidium (GA) into the perovskite bulk while simultaneously using 2-(4-methoxyphenyl) ethylamine hydrobromide (CH_3_O-PEABr) for surface passivation [30]. The incorporation of GA into the bulk mixed organic perovskite formed a 2D/3D heterostructure of the FAGAPbI_4_ phase, where 2D nanosheets were observed at the perovskite surface. This morphology alters the electrical properties of the bulk by reducing the trap-induced charge nonradiative recombination center. Surface treatment of CH_3_O-PEABr further enhanced the performance due to the decrease in surface roughness, as it was more favorable for successive layers of 2,2′,7,7′-tetrakis [N,N-di(4-methoxyphenyl)amino]-9,9′-spirobifluorene (Spiro-OMeTAD) HTL for better charge transport between the perovskite/HTL interface. Lead thiocyanate (Pb(SCN)_2_) additives were used in PPVs [85,86,87] to increase the grain size, improve crystallinity, and produce excess PbI_2_, which is beneficial for grain boundary passivation. The same method was employed for Id-PPVs by adding Pb(SCN)_2_ as a bulk additive and PEABr for surface modification [32]. Similarly, Pb(SCN)_2_ additives increased the grain size and crystallinity, and the PEABr surface treatment further improved the photoelectrical properties by passivating defects at the perovskite surface. Remarkably, a high PCE of 37.18% with a V_oc_ of 1.12 V was obtained under 1000 lx of weak LED illumination.

Additionally, bilateral inorganic “walls” of alkali fluoride, including LiF, NaF, and KF, sandwiched the perovskite layer under both ETL and HTL to effectively decrease the trap-assisted recombination and avoid bidirectional ion migration at the interface [54], as shown in Figure 3c. They hypothesized that KF-treated samples performed better than other alkali fluorides, because K has the smallest electronegativity, and hence, K-I bonds are preferred over Li-I and Na-I at the interface, leading to stronger interactions between perovskite and the AF layer, as shown in Figure 3d. Consequently, 35.7% of PCE was obtained under 1000 lx LED illumination compared to only 24.9% PCE for the control samples.

Similarly, a buffer layer between the CTL and metal electrode is needed because it can prevent metal diffusion to the CTL layer, which is related to the rapid retardation of the PPV performance. Li et al. were the first to study the addition of an ionic liquid as a buffer layer to inverted Id-PPVs between the ETL and electrode [56]. Not only does it enhance the electrical properties but also protects the devices from the diffusion of moisture and oxygen into the perovskite layer, which is very prone to them. Further research on interface engineering for Id-PPVs applications should be explored, as it may provide greater potential to achieve its best performance.

### 3.4. Other Strategies

Inorganic perovskite is considered a good candidate to obtain the maximum performance of Id-PPVs, as it has a matched bandgap to the common indoor lamp spectra, which is approximately 2 eV. Additionally, a highly crystalline inorganic perovskite is required to achieve a higher performance. Ghosh et al. studied the engineering of crystalline grains of solution-processed Cs-based perovskites by employing different types of antisolvents, yielding 14.1% of PCE under 1000 lx of white LED [88]. The antisolvent treatment strongly affected the density and compactness of the Cs-based perovskite and the coverage of the surface. This was correlated with the ability of an antisolvent to reduce the defect concentration and its capability to suppress grain misorientation. Other studies on PPVs also revealed that different types of antisolvent could be categorized based on their ability to dissolve organic precursors and their miscibility with host perovskite solution solvents [89]. Specifically, there is still much room for improvement and optimization to maximize the potential for indoor light energy conversion.

## 4. Towards Id-PPVs Commercialization

### 4.1. Human Friendly Id-PPVs

PPVs incorporating lead (Pb) still exhibit the highest performance [21,22]. However, Pb, the major component in the perovskite structure, is one of the obstacles to the commercialization of Id-PPVs because of their toxicity, considering that Id-PPVs are in close proximity to humans. Recently, Pb-free perovskite materials were widely investigated as alternative materials to Pb. Tin (Sn) is a promising candidate for replacing Pb in perovskite structures. Sn-based perovskites have a suitable bandgap for solar cells, but they exhibit poor stability due to the oxidation of Sn^2+^ to Sn^4+^ [90]. Yang et al. incorporated an abundant human-friendly material in nature, called Catechin, with a hydroxyl functional group, as a doping agent into Sn-based Id-PPVs to suppress Sn oxidation [91], as shown in Figure 4a. It also showed an improvement in film morphology due to slowed crystallization. Subsequently, a PCE of 12.81% with 0.65 V of V_oc_ was obtained compared to 10.45% for those of the control devices under 1000 lx illumination. The highest PCE of Sn-based PPVs to date was attained by Cao et al. on the bottom passivation of an ETL/perovskite interlayer using multifunctional KSCN materials [92], as shown in Figure 4b. A PCE as high as 17.57% was obtained compared to only 13.57% for those of the control devices under 1062 lx of LED lamp illumination, which proved that K^+^ and SCN^−^ ions diffused from the interface into the perovskite bulk and passivated defects both at the interface and in the bulk. The SCN^−^ ion inclusion depressed Sn^4+^ oxidation, leading to better stability, where it was tested at ambient temperature without illumination inside a glovebox. After 1200 h, it retained 80% of its original PCE, whereas only 60% was retained in the control devices. Furthermore, antimony-based [93,94] and bismuth-based perovskite [95], with 1.9 and 1.78 eV bandgap, respectively, have been studied as alternatives to the Pb-based Id-PPV. However, it is still challenging to improve Pb-free perovskite-based devices. Further exploration of highly efficient Pb-free Id-PPVs is required.

The PCE of Pb-free devices has a large technical gap compared to Pb-based devices. Thus, another strategy is to prevent any Pb leakage from Pb-based Id-PPVs through encapsulation. There are various encapsulation materials and techniques, such as organic polymers with hydrophobicity and flexibility [96,97,98], inorganic materials with high stability from environmental stimuli [99,100,101], carbon-based materials exhibiting good optical and mechanical properties [102], multi-layer/composition of hybrid encapsulation [103,104,105,106,107], and Pb absorbents within PPVs [108,109,110]. However, materials and structures for advanced encapsulation still need further development to prevent Pb leakage during practical use. Additionally, tremendous efforts were made to overcome the Pb issue in PPVs, which is already discussed in earlier studies [111,112,113,114], although it is still challenging to improve Pb-free perovskite-based devices, especially for indoor applications. Further exploration of highly efficient Pb-free Id-PPVs is required.

### 4.2. Flexible Id-PPVs

Perovskite materials with excellent mechanical durability were proven to be applicable on flexible substrates that provide vast integration, such as biosensors and rollable flexible PPVs (f-PPVs) [29]. Moreover, flexible thin glass, indium tin oxide (ITO)-coated polyethylene naphthalate (PEN), and polyethylene terephthalate (PET) polymer are commonly used as transparent flexible substrates. These substrates have lower light transmittance than their glass counterparts, resulting in poorer device performance [115]. Therefore, understanding the behavior of photoelectric conversion under lowlight illumination on flexible substrates is important considering its potential. The study was begun by Castro-Hermosa et al. with the implementation of Id-PPVs on flexible thin glass (approximately 100 μm) manufactured via a roll-to-roll sputtering procedure [116], as shown in Figure 5a. This method is preferred for large-area manufacturing and mass production because it exploits the benefits of solution processing and higher output per unit time. A PCE of 22.6% (35.0 μWcm^−2^ power output) was obtained under 400 lx of LED illumination.

**Figure 4 nanomaterials-13-00259-f004:**
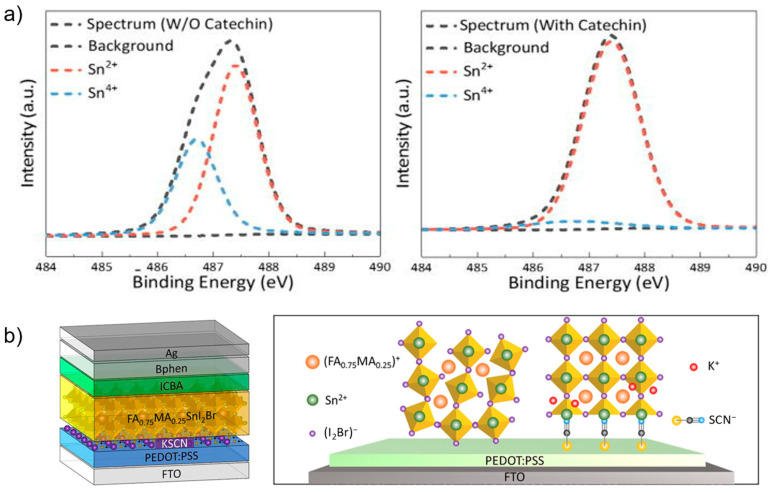
(**a**) Plots illustrating the Sn^4+^ suppression upon Catechin addition from X-ray photoelectron spectroscopy measurement. Reprinted with permission from [91], copyright 2021 AIP Publishing. (**b**) Schematic illustration of the Sn-based PPVs with KSCN treatment and of crystal orientation of FA_0.75_MA_0.25_SnI_2_Br from (0 0 1) plane, with and without KSCN treatment. Reprinted with permission from [92], copyright 2022 Elsevier.

Crosslinking materials are used in Id-PPVs to maximize the full-dimensional stress release [60] and device endurance over moisture and oxygen [61]. A 31.85% PCE was obtained under 1062 lx LED illumination and maintained 91.8% of its initial PCE after 10,000 cycles of a bending test of 5 mm [60]. The latter obtained 30.73% PCE under 1000 lx of white LED and sustained more than 80% of its initial PCE over humidity and lighting stability tests. Fabricating perovskite films on flexible substrates is difficult because of their wavy nature. Therefore, the air gap between the glass fixture and flexible substrate was removed by employing a van der Waals stacking process based on polydimethylsiloxane coating for conformal heat transfer throughout the film [50], as shown in Figure 5b. With an additional vacuum-based perovskite post-treatment using oleyl amine, a very intriguing result of more than 40% PCE was obtained under 1000 lx of cool white LED illumination. These results imply that obtaining higher-performance Id-PPVs on flexible substrates remains a challenge to be solved.

### 4.3. Large Area Id-PPVs

For real-life applications, it is essential to fabricate Id-PPVs that are sufficiently large to supply energy for various applications. The estimated Id-PPV area required for radio-frequency identification (RFID), passive Wi-Fi, sensors, and stand-still camera under 1000 lx with ~21% PCE (power density output of ~80 μWcm^−2^) is approximately 0.2, 0.3, 0.9, 1.3, and 16.5 cm^2^, respectively [116]. The main problem is that perovskite nucleation and crystal growth become more difficult to control as the active area increases. Table 2 summarizes the studies on larger area Id-PPVs.

Vacuum-based deposition of a perovskite layer fabricated over a 5.68 cm^2^ substrate yielded a PCE of 24.9% (77.6 μWcm^−2^) [63]. The reduced defect density facilitated the higher performance of the C_60_ ETL compared to that of the spin-coating method. Unfortunately, this method is not favorable for commercialization because of the high vacuum required during deposition; thus, other methods, such as roll-to-roll [116] or blade coating [57] are preferred. Blade-coated devices exhibited 33.8% PCE for 1 cm^2^ devices under 1000 lx of white LED illumination and were further able to power up a LED by connecting some devices in series with a total active area of 4 cm^2^, as shown in Figure 5c,d. The perovskite film deposited on an overall 9 cm^2^ active area showed a PCE of 34.86% under 1000 lx of fluorescent lamp and 35.7% under 1000 lx of white LED lamp, applying lead oxalate as perovskite additives [49] and alkali fluoride as perovskite top-bottom passivation [54], respectively.

**Figure 5 nanomaterials-13-00259-f005:**
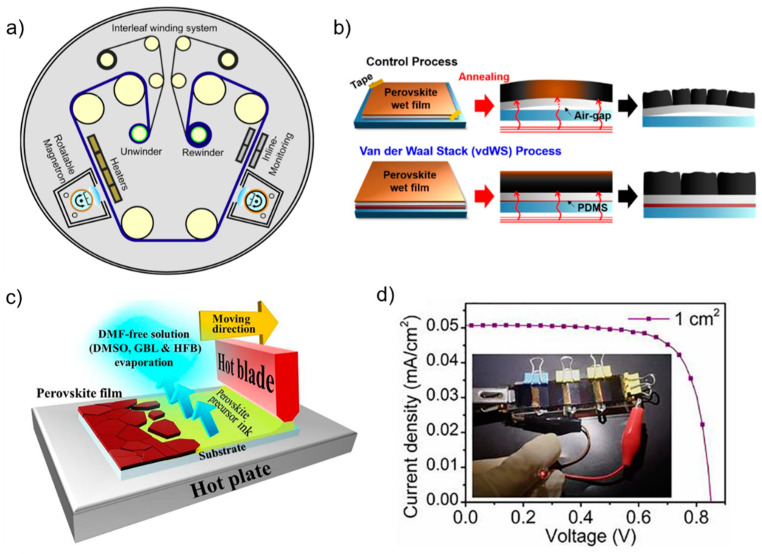
(**a**) Schematic illustration of the roll-to-roll equipment via rotatable magnetron sputtering on a 100 μm-thick glass roll. Reprinted with permission from [116], copyright 2020 Elsevier. (**b**) Schematic illustration of annealing behavior and crystallization formation with and without air gap, and J–V curve under 1000 lx illumination of LED with inset stabilized PCE. Reprinted with permission from [50], copyright 2022 American Chemical Society. (**c**) Schematic illustration of the one-step blade-coating method assisted by using hexafluorobenzene as co-solvent. (**d**) Plot of the J–V curve of series-connected mini-module under white LED 285 lux illumination. Reprinted with permission from [57], copyright 2022 Elsevier.

To further exploit the full potential of Id-PPVs, a module design with many sub-cells connected in series was developed to reduce the series resistance, which may increase scalability. Mathews et al. built an Id-PPV fully powered device called RFID temperature sensor, which consisted of three cells with 0.7 cm^2^ active area, as defined from gold contact pads [117]. It produced a power output as high as 14.5 μW at 1.81 eV bandgap, with an exceptional 13.2% PCE under 0.16 mWcm^−2^ of fluorescent illumination. Unfortunately, studies on module characteristics under low-light illumination are still lagging, compared to those on the comprehensive explanation of module design for PPVs [118,119]. Thus, further studies are required to fill the existing research gap.

## 5. Conclusions

Perovskite absorber with an optimized bandgap of ~1.7–2.0 eV is required to utilize the full potential of Id-PPVs as power suppliers or battery rechargers for indoor IoT systems. This can be realized through halide composition engineering. In fact, halide composition engineering can not only alter the bandgap but also improve the film quality upon transformation of the crystal structure, leading to a better morphology. Furthermore, various strategies, including doping and post-treatment of the CTL layer, can tune its physical morphology and optoelectrical properties, which are favorable for carrier separation and transport. Hence, comprehensive studies employing polymeric molecules, ionic liquids, self-assembled monolayers, or carbon-based materials for CTL modification under indoor light should be conducted. Furthermore, hydrophobic materials may further increase the device performance and stability by controlling ion migration out of the perovskite film and preventing water contamination into it, which can lead to degradation.

Id-PPVs for IoT system integration can be implemented in offices, schools, supermarkets, and other places that are in close proximity to humans. Pb, as the main component of perovskite materials, is still the most threatening issue for commercialization because of its toxicity to the human body and plants. Therefore, Pb leakage must be prevented through the development of technologies, such as advanced encapsulation, Pb-free halide perovskite, and the recycling process.

To apply Id-PPVs to various electronics, it is essential to implement various form factors and achieve high performance. Among photovoltaics, PPVs already exhibited the highest specific power per weight (W/g) and excellent mechanical properties rendering them foldable, crumbly, and stretchable. From these advantages, there are various electronic applications, such as temperature sensors [117], integrated circuits with LEDs [63], e-paper [120], unmanned aerial vehicles [121], and wearable solar clothing [29]. However, it remains challenging to improve the stability of flexible and lightweight PPVs under extreme environmental conditions such as humidity, stress, and scratching. Although encapsulants can protect devices from ambient conditions, environmental shock can break the encapsulation layer outside the PPVs. For practical use, a multifunctional encapsulation layer should be further developed to provide multifunctional properties such as watDeerproofing, flexibility, and high-impact strength.

## Figures and Tables

**Figure 3 nanomaterials-13-00259-f003:**
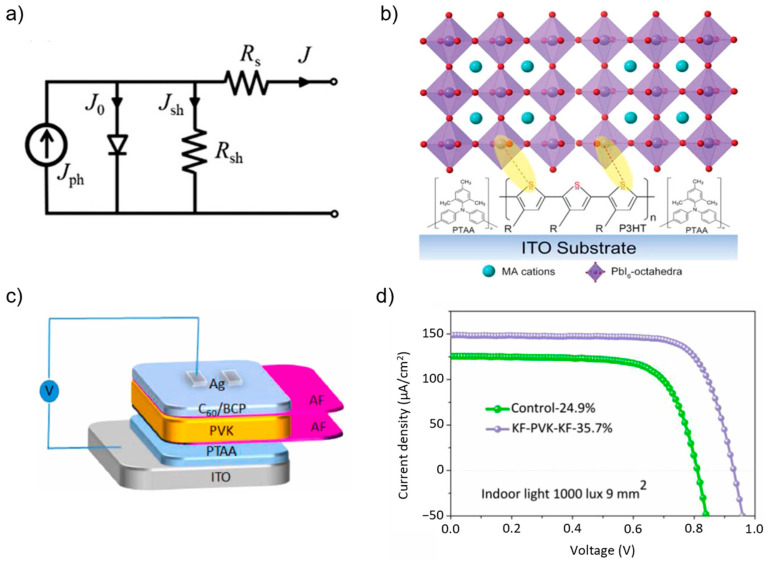
(**a**) Diagram of the equivalent circuit of solar cell, with the basic parameters defined as the reverse saturated current density (J_0_), shunt current (J_ph_), photogenerated current (J_ph_), measured current (J), shunt resistance (R_sh_), and series resistance (R_s_). Reprinted with permission from [62], copyright 2021 John Wiley and Sons. (**b**) Schematic illustration of the interaction between hydrophobic poly(3-hexylthiophene) (P3HT) molecule and perovskite single crystal. Reprinted with permission from [37], copyright 2021 John Wiley and Sons. (**c**) Schematic illustration of the PPV device structure with alkali fluoride top-bottom passivation. (**d**) Plot of the J–V curve of a 9 mm^2^ device at 1000 lx. Reprinted with permission from [54], copyright 2021 Elsevier.

**Table 2 nanomaterials-13-00259-t002:** Summary of large area Id-perovskite photovoltaics (PPVs) under 1000 lx illumination and its output performance.

Substrate	Key Technology	Lamp Type	Area (cm^2^)	Power Output (μWcm^−2^)	PCE (%)	Ref
FTO	Ionic liquid for ETL modification	FL	9	-	35.2	[56]
ITO	Non-halide perovskite additives	FL	9	-	34.86	[49]
ITO	NiO-based HTL	CWLED	1	82.12	27.43	[58]
ITO	Alkali fluoride perovskite passivation	WLED	9	-	35.7	[54]
FTO	Blade coating	WLED	1	-	33.8	[57]
PET/ITO	Polymer-based perovskite passivation	WLED	24	-	30.73	[61]
ITO	Halide for bandgap engineering	FL and LED	1	-	17.89	[40]
ITO	Vacuum-based coating	FL	5.68	77.6	24.9	[63]

CWLED, cool white light emitting diode; ETL, electron transport layer; FL, fluorescent lamp; FTO, fluorine-doped tin oxide; HTL, hole transport layer; ITO, indium tin oxide; PET, polyethylene terephthalate; WLED, white light emitting diode.

## Data Availability

Not applicable.

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
