# Peer review of "Recent Strategies for High-Performing Indoor Perovskite Photovoltaics"

_nanomaterials, 2023, doi:10.3390/nano13020259_

Round 1

Reviewer 1 Report

Review for nanomaterials-2112692

This paper reviews the research progress on perovskite photovoltaics (PPVs) under indoor light conditions, with a focus on device engineering to achieve high-performance in-door PPVs (Id-PPVs), including bandgap optimization and defect management. In the end, the authors discussed the challenges of Id-PPVs development and its interpretation as a potential research direction in the field.

The review think the main text, all the schemes, figures and tables basically have been carefully prepared. Therefore, I recommend publication of the manuscript. However, some minor errors and mistakes should be considered and resolved in the final revision.

1. Line 152, 161,163, 334, 335: for C2O4-, HSO4-, SCN-, - should superscript.

2. The couclusion section may be much longer. Please reprepare a new shorter and conise one.

3. Line 216: the authors mentioned “polymer-based flexible substrates [49]”. The supported reference may be not enough. And the following references may be cited to support this opinion.

[1] J. Solid State Chem. 313 (2022) 123271. DOI: 10.1016/j.jssc.2022.123271.

[2] Inorg. Chim. Acta 530 (2022) 120697. DOI:10.1016/j.ica.2021.120697.

[3] Z. Naturforsch. B Chem. Sci. 2022, 77, 561-564. DOI: 10.1515/znb-2022-0031.

[4] Inorg. Chem. 2021, 60, 10109-10113. DOI:10.1021/acs.inorgchem.1c01541.

[5] J. Solid State Chem. 305 (2022) 122636. DOI: 10.1016/j.jssc.2021.122636.

[6] New J. Chem. 41 (2017) 12611–12616. DOI:10.1039/C7NJ02021J.

4. The references section: please give the correct subscript numbers for the molecular formula of the correspondig compounds, involving in Refs [33, 41, 43, 44, 47].

5. The references section: reference format is not uniform, some journal name are abbreviation, some are not. Please refine.

Reviewer 2 Report

1. As an indoor perovskite photovoltaics technology, toxic lead in perovskite is so closely exposed to humans. Therefore, the latest research progress of lead leakage prevention and lead-free perovskites should be summarized in detail in this review.
2. As a review, each chapter should have a summative conclusions or perspectives.
3. It is mentioned in the article that TiO2 is used as a transport layer in some studies. What are its advantages or disadvantages compared with meso-TiO2 and c-TiO2?
4. How does the low-temperature solution treatment mentioned in Section 3.3.2 improve the properties of HTL?
5. With regard to human friendly Id-PPVs, this article only summarizes the progress of Sn-based Id-PPVs. Can you briefly introduce other non lead based Id-PPVs with bandgap matching?
6.  On lines 116 and 117, the symbol for the radius unit angstrom should be Å instead of A.
7. On lines 158-169, The “Du et al.” and “[41]” are repeated.
8. On line 181, “have a” should be changed to “have”.
9. On line 447, “Of the other photovoltaics” should be changed to “In the other photovoltaics”.

Reviewer 3 Report

This manuscript studies the indoor perovskite photovoltaics (Id-PPVs) which is the best candidates for self-powered IoT system integration. The novelty of the study lies in the efforts made thus far in the field of Id-PPVs, including recent strategies to achieve high-performance Id-PPVs, the development of human-friendly Id-PPVs, and other key aspects for commercialization. This work is well worth to be accepted for publishing in the MDPI Nanomaterials and could reach a wide range of audience, but there are still several areas that need to be revised and improved.

1.     The introduction begins with an overview of IoT and why it is important nowadays. It then continues with the features of IoT sensors and a problem that might occur in the system. Unfortunately, the explanation of the features and its problem are insufficient to the latter idea, which is the solution of the problems, including different types of energy harvesting devices for wireless and self-sustainable operation IoT sensors. Thus, more comprehensive writing needed to make it clearer as the basic reason why indoor photovoltaics is the best candidates among others.

2.     The authors also stated in the introduction that Id-PPVs are the best choice among others because it exhibits higher light to power conversion efficiency, but there is no clear comparison between them specifically. I strongly suggest elaborating on this section and providing appropriate recent references. Ex.) Energy Environ. Sci. 2022, 15, 2479.; ACS Energy Lett. 2022, 7, 2547.; Matter 2022, 5, 725  for organic Id-PV.

3.     The authors suggest several requirements such as high-performing, human-friendly, and flexible large-area Id-PPVs for commercialization and vast application. Unfortunately, there is a lack of exposure to the successful integration of Id-PPVs and other electronic devices. The authors need to add some real applications of Id-PPVs to further emphasize the importance of Id-PPVs.

Reviewer 4 Report

The manuscript "Recent strategies for high-performing indoor perovskite photovoltaics" is well written, thorough and organized. For these reasons, I suggest its publication in Nanomaterials, after some minor revisions. 

- Page 2, line 51: It is stated that perovskite solar cells have low prodution costs. However, it is true until gold is not used as counterelectrode and the production is carried out under ambient conditions. Gold can be repleased by carbon-based materials (as reported in Energy Environ. Sci. 2019, 12, 3437-3472). Ambient condition processes can be carried out as well (as reported in Solar RRL, 2021, 5, 2100341 and Solar Energy 2021, 224, 1369-1395). I kindly suggest to include these considerations and references. 

- Page 2, line 86: 'et al.' in italics.

- Page 3, line 97: 'et al.' in italics.

- Page 3, line 107 (caption Fig 1): 'et al.' in italics.

- Page 4, line 148: 'et al.' in italics.

- Page 4, line 149: 'et al.' in italics.

- Page 4, line 148: 'Du et al.' is repeated twice. 

- Page 5, line 170: 'et al.' in italics.

- Page 5, line 202: 'et al.' in italics.

- Page 5, line 214: 'et al.' in italics.

- Page 6, line 219: 'et al.' in italics.

- Page 6, line 226 (caption Fig 3a): Please, specify the physical properties (such as Rs, Rsh), indicated in the picture. 

- Page 7, line 247: 'et al.' in italics.

- Page 7, line 262: The 40.1% PCE taken from ref 16 is above the Shockley-Queisser limit. Please, specify how the authors of the work overcome this limit. 

- Page 7, line 291: 'et al.' in italics.

- Page 8, line 307: 'et al.' in italics.

- Page 9, line 325: 'et al.' in italics.

- Page 9, lines 325-327: in ref. 71, Catechin is incorporated into the cell to suppress Sn oxidation. Please, specify how this material is incorporated. 

- Page 9, line 349: 'used as transparent flexible'.

- Page 9, line 353: 'et al.' in talics. 

- Page 10, line 391: 'to power up a LED'

- Page 11, line 413: 'et al.' in italics.

Round 2

Reviewer 2 Report

no